nanotechnology/inorganic chemistry/materials science

lattice constant, strain, scanning electron microscopy, band gap, X-ray diffractometer

**Author for correspondence:**
S. A. Saah
e-mail: selina.saah@uenr.edu.gh

This article has been edited by the Royal Society of Chemistry, including the commissioning, peer review process and editorial aspects up to the point of acceptance.

# Lead ethyl dithiocarbamates: efficient single-source precursors to PbS nanocubes

S. A. Saah[1], N. O. Boadi[2], D. Adu-Poku[1] and C. Wilkins[3]

[1]Department of Chemical Sciences, University of Energy and Natural Resources, Sunyani, Ghana
[2]Department of Chemistry, Kwame Nkrumah University of Science and Technology, Kumasi, Ghana
[3]School of Materials, The University of Manchester, Oxford Road, Manchester M13 9PL, UK

SAS, 0000-0002-0585-2144

Lead ethyl dithiocarbamates have been successfully used as single-source precursors for the deposition of PbS using spin coating followed by annealing at moderate temperatures. The thin films were characterized using a powder X-ray diffractometer and were found to be face-centred cubic with the (200) plane being the most preferred orientation. Scanning electron microscopy images showed the formation of well-defined cubes. Optical band gaps of PbS thin films were estimated using Tauc plots as 0.72, 0.73 and 0.77 eV at annealing temperatures of 250, 300 and 400°C. These band gaps were all blue shifted from the bulk value of 0.41 eV. Energy-dispersive X-ray analysis was used to determine the composition of the thin films which showed an approximately 1 : 1 Pb to S ratio.

## 1. Introduction

Single-source precursors (SSPs) have been widely studied recently due to their efficiency as starting materials to the syntheses of nanomaterials [1–4]. Common examples include carbamates [5], xanthates [6], urea [7], imidodiphosphinates [8], phosphinates [9] and biurets [10]. These SSPs have multiple advantages over their dual-source counterparts as reviewed earlier in the literature [11,12].

Dithiocarbamates are highly versatile ligands that form stable complexes with most metal ions [5,13]. Aside from their use as starting materials for the syntheses of nanomaterials, they have been reported as antimicrobial agents [14], herbicides [15], insecticides [16] and flotation agents [17]. Extensive research has been conducted on metal dithiocarbamate complexes due to their strong metal-binding properties. This is as a result of the presence of two-electron donor sulfur atoms which determines the stability of the resulting complex [18]. Dithiocarbamates are soft sulfur donor ligands, and the O'Brien group was the first to report on the use of lead(II) dithiocarbamato complexes [Pb(S$_2$CNRR')$_2$] (R, R' = ethyl, butyl, $^i$butyl) as SSPs for the syntheses of PbS nanoparticles [19]. Using trioctylphosphine oxide (TOPO) as

capping agents, the researchers synthesized nanocrystalline PbS by thermolyses of the SSPs. It was observed that the optical and morphological properties of the PbS nanocrystallites depended strongly on the temperature rather than the chemical nature of the precursors. For example, at 100°C, spherical PbS nanocrystallites with average diameters of 6.3 nm were obtained, whereas a mixture of cubic and spherical crystallites was obtained at 150°C.

PbS nanomaterials have been synthesized using several techniques; however, the spin coating of SSPs onto substrates followed by annealing at moderate temperatures offers a simple, quick, cost-effective and industrially scalable route for the production of high-quality thin films [6,20,21]. Depending on the reaction conditions, several morphologies have been reported [22,23]. These include cubes [24], rods [3], octahedron [25], rod with cube at the tip [20], rod interdispersed with cubes [21], spheres [26] mesh-like structures [27], stars [28], pyramid [29], wires [30] and dendrites [31]. PbS is a group IV–VI semiconductor which possesses much higher dielectric constants ($\varepsilon$) and lower effective mass for the electron and the hole [32]. It is a direct band gap semiconductor with an exciton Bohr radius of 18 nm and a narrow band gap of 0.41 eV [21]. PbS can be size-tuned to absorb strongly over a wide range of wavelength on the electromagnetic spectrum thereby shifting its absorption edge into the near infrared region [33]. PbS has widely been used in several opto-electronic devices which include photovoltaic cells [34], sensors [35], thermoelectrics [36], photodetectors [37], diodes [38], catalyst [39], photoconductors [40] and solar concentrators [41]. We have previously reported the syntheses of PbS nanoparticles and thin films using xanthates as precursors [6,21]. There is, however, no report on the deposition of PbS thin films using lead diethyldithiocarbamates SSPs using the spin coating method.

Herein, this report outlines a simple and straightforward synthesis of lead ethyl dithiocarbamates and their use as starting materials for the deposition of PbS thin films.

# 2. Experimental

## 2.1. Materials

Lead acetate trihydrate 99%, toluene 99.8%, chloroform 98% and sodium diethyldithiocarbamate 98% were used as received from Sigma Aldrich.

## 2.2. Instrumentation

Elemental analyses (CHNS) were carried out on Flash 2000 Thermo Scientific elemental analyser and TGA data obtained with Mettler Toledo TGA/DSC1 Star System between the ranges of 30–600°C at a heating rate of 10°C min$^{-1}$ under nitrogen flow. Scanning electron microscopy (SEM) and energy-dispersive X-ray (EDX) spectroscopy were carried out using a Philips XL 30 FEG scanning electron microscope equipped with a DX4 EDX detector and was used to determine surface morphology and elemental composition of the nanoparticles. All samples were carbon coated using Edwards coating system E306A prior to SEM and EDX analyses. Powder X-ray diffraction (p-XRD) analyses were done using a Bruker AXS D8 diffractometer employing CuK$\alpha$ radiation ($\lambda = 1.5418$ Å) at 40 kV and 40 mA at room temperature. The PbS nanoparticles were scanned between 20° and 90° with a step size of 0.02° and dwell time of 3 s. Electronic absorption measurements were performed on Perkin Elmer UV–VIS–NIR lambda 1050 double beam spectrophotometer.

## 2.3. Synthesis and characterization of lead ethyl dithiocarbamate complex

The lead ethyl dithiocarbamate complex was synthesized as reported elsewhere in the literature [19]. Typically, 20 mmol of the sodium ethyl carbamate was stirred in distilled water (50 ml) until complete dissolution. Lead acetate trihydrate (10 mmol) was dissolved in distilled water (50 ml) and added dropwise to the sodium ethyl carbamate solution. The resulting precipitate was filtered, washed three times with distilled water (50 ml) and air dried. The crude complex was recrystallized in toluene to give crystalline yellow lead ethyl dithiocarbamate complex. Yield = 92%. The melting point is 210–211°C. Micro-elemental analyses: calc (found); C, 26.90 (26.85); H, 5.27 (5.26); N, 5.23 (5.22); Pb, 38.67 (38.65) and S, 23.94 (23.90). $v_{(C–S)}$ 1265 cm$^{-1}$, $v_{(C–N)}$ and 1133 cm$^{-1}$. $^1$HNMR (CDCl$_3$, 400 MHz) $\delta$/ppm: 1.26 (t, $J = 7.2$ Hz, 3H; CH$_3$), 3.67 (q, $J = 7.1$ Hz, 2H; CH$_3$).

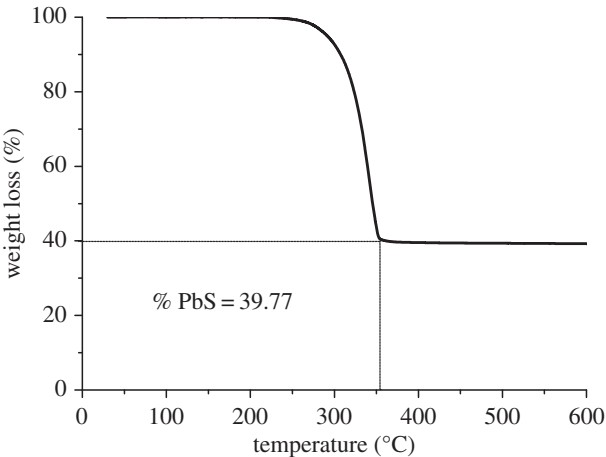

**Figure 1.** TGA thermograph of lead diethyldithiocarbamate complex.

## 2.4. Deposition of PbS thin films

Precursor solutions were prepared by dissolving lead ethyl carbamate (0.2 g) in chloroform (2 ml) and spin coating at 1500 r.p.m. for 20 s onto glass substrates. The coated glass substrates were heated at 200, 250, 300 and 400°C for 30 min under nitrogen gas.

## 3. Results and discussion

The lead diethyldithiocarbamate complex was synthesized by exchanging the sodium atom in the ligand with the lead atom. The use of water as a solvent makes the reaction process environmental friendly. Water as a reaction medium has been reported to give high yields when compared with other organic solvents such as methanol, chloroform and hexane [42]. Micro-elemental analysis of the complex is in good agreement with the proposed formula further confirming its high purity. An ideal complex that is suitable as a starting material for the syntheses of nanomaterials should have specific characteristics. These include satisfactory solubility in common organic solvent and its ability to stay undecomposed upon exposure to moisture and air at room temperature [20]. The lead diethyldithiocarbamate complex possessed these characteristics.

The FTIR spectrum of the lead diethyldithiocarbamate complex showed characteristic absorptions at 2967 and 2928 cm$^{-1}$ which could be attributed to CH antisymmetric and symmetric stretches, respectively. Other equally important stretches such as C–S and C–N occurred at 1265 and 1133 cm$^{-1}$, respectively, as reported in the literature [39,43]. The single absorption peak around 981 cm$^{-1}$ indicates the coordination of the diethyldithiocarbamate ligand, in a bidentate manner, to the lead ion [39].

The complex decomposed in a single step to yield a stable residue which is 39.77% of the initial mass of sample used (figure 1). The onset and offset decomposition temperatures were at 216 and 350°C, respectively. The 100% conversion of the complex to PbS results as 44.60% of the total mass. The difference, however, between the two values is 4.90% which implies that the compound can be decomposed to produce PbS as residue [21].

Each of the p-XRD patterns of as-deposited PbS thin films was indexed as the face-centred cubic (fcc) phase of PbS (galena) with (111), (200), (220), (311), (222), (400), (311), (420), (422) and (511) as the faces (ICDD 00-003-0614). Generally, there were eight distinct diffraction peaks in the XRD pattern of the PbS obtained at 250, 300 and 400°C. The purity of the PbS produced was highly dependent on the annealing temperature (figure 2). At 200°C, the other peaks in the pattern primarily were as a result of undecomposed precursor which did not match to any phase of PbS. An increase in temperature to 250°C showed the complete decomposition of the precursor to a pure cubic phase PbS with no additional peaks from either the precursor or other phases of PbS. Similar pure PbS phase spectra were obtained at 300 and 400°C confirming the purity of the products [44]. Similar observations on the effect of deposition temperature on the *in situ* thermal decomposition of lead(II) *n*-octylxanthate within a 1,3-diisopropenylbenzene–bisphenol A dimethacrylate sulfur copolymer has been reported [45].

Structural parameters such as crystallite size (*D*), dislocation density (*δ*), lattice constant (*a*) and strain (*ε*) were calculated from the p-XRD data [46].

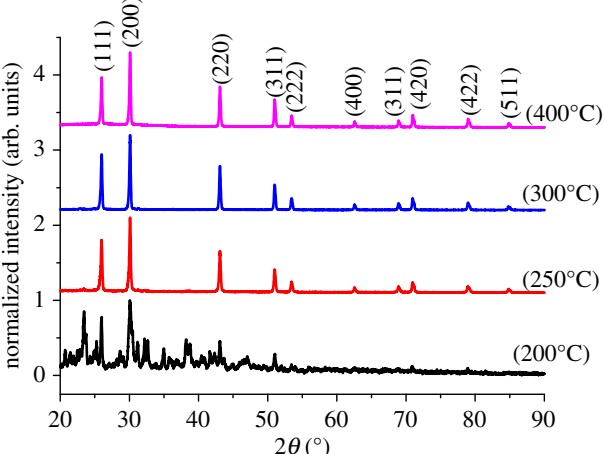

**Figure 2.** p-XRD pattern of PbS thin films at 200, 250, 300 and 400°C thermal decomposition temperatures.

**Table 1.** Structural analyses from p-XRD data.

| temperature (°C) | crystallite size (nm) | size from WH plot (nm) | dislocation density | lattice constant (Å) | strain |
|---|---|---|---|---|---|
| 250 | 34.8 | 50.0 | 0.00082669 | 5.93439 | 0.0010 |
| 300 | 34.9 | 51.28 | 0.00082101 | 5.93375 | 0.0008 |
| 400 | 37.5 | 48.08 | 0.00071149 | 5.93272 | 0.0007 |

The crystallite size calculated from the Scherrer equation (eqn. (3.1)) ranged from 34.78 to 37.49 nm as the temperature increased from 250 to 400°C (table 1). There was no significant change in the width of the peaks and therefore there was not much significant change in the calculated crystallite sizes.

$$D = \frac{k\lambda}{B\cos\theta}, \tag{3.1}$$

where $D$ is the crystallite size (nanometre), $k$ is a dimensionless shape factor, with a typical value of about 0.9, $\lambda$ is the wavelength of the radiation, $B$ is the full width at half maximum (FWHM) and $\theta$ is the angle of diffraction (Bragg angle).

There was a decrease in the number of defects, which represents the dislocation density as the crystallite sizes increased (table 1). This may be due to a decrease in the occurrence of grain boundaries as the crystallite size increases with temperature [47,48]. The dislocation density was calculated from Williamson and Smallman's equation,

$$\delta = \frac{1}{D^2}, \tag{3.2}$$

where $D$ is the average crystallite size.

The lattice constant $a$ refers to the physical dimensions of the unit cell in a crystal lattice [21]. Usually, lattices are represented by three constants $a$, $b$ and $c$ which are referred to as lattice parameters. However, in a special case of a cubic crystal, the three lattices are equal and usually referred to as $a$. The calculated lattice constant ranged between 5.9327 and 5.9344 Å, which was similar to that of bulk PbS, which is 5.9362 Å [11]. The lattice constant for the cubic phase is determined by using the equation below:

$$a = d\sqrt{(h^2 + k^2 + l^2)}, \tag{3.3}$$

where $d$ is the spacing between the planes in the atomic lattice, and $h$, $k$ and $l$ are the Miller indices.

The Williamson–Hall (WH) plot is a useful tool for graphically demonstrating the $hkl$-dependence of broadening within a diffraction pattern [49]. A plot of $\beta\cos\theta/\lambda$ against $4\sin\theta/\lambda$, which is the Williamson–Hall plot, gives a linear plot with strain as the slope and particle size as the inverse of the intercept (electronic supplementary material, figure S1) [50]. The positive slope in the Williamson–Hall plot implies that tensile strain was dominant in all the PbS thin films [51]. The microstrain developed in

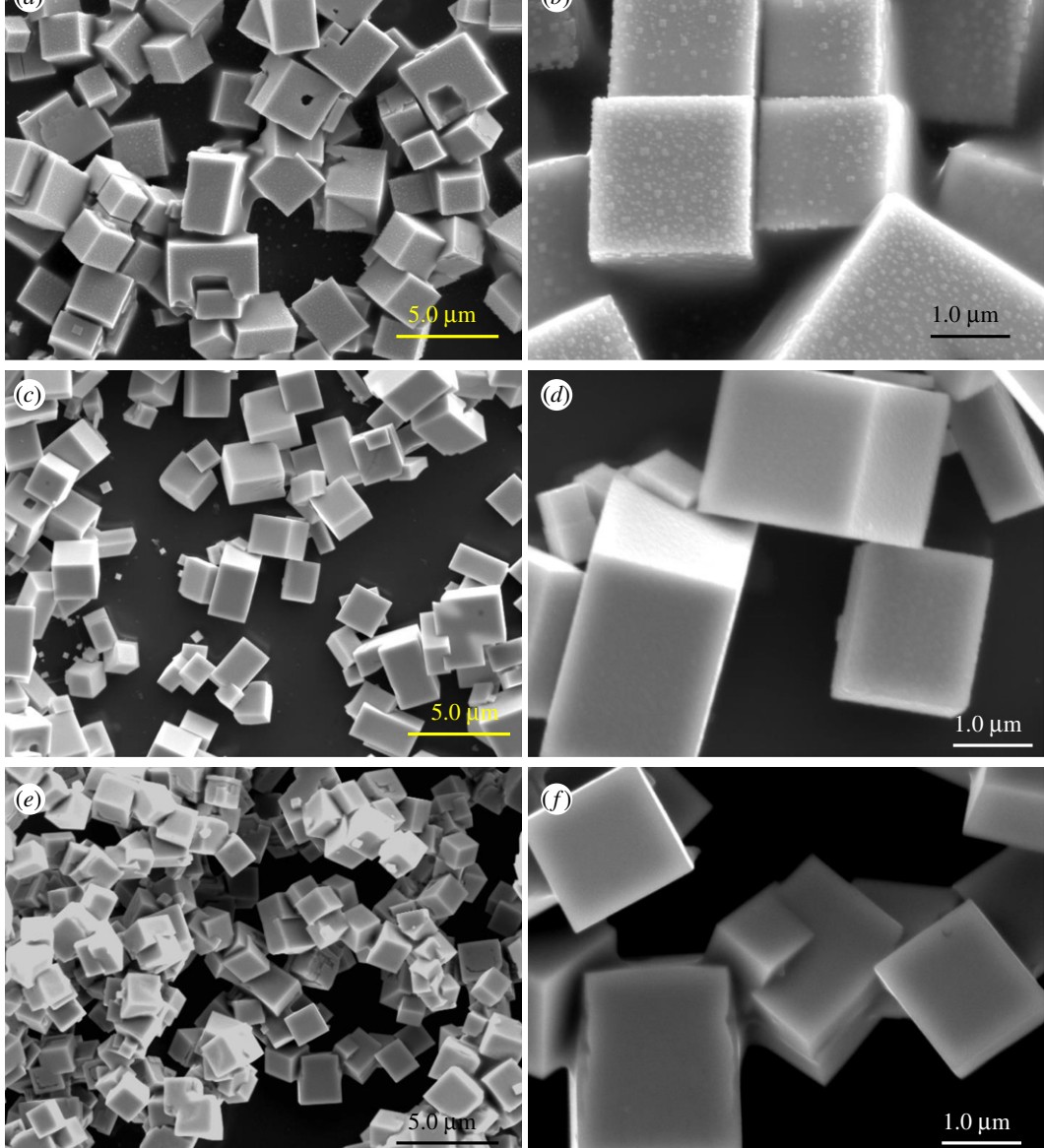

**Figure 3.** SEM images of all PbS thin films taken at various magnifications (instrument magnifications of 5000× and 20 000×) to illustrate the overall morphology of PbS thin films obtained at 250°C (*a,b*), 300°C (*c,d*) and 400°C (*e,f*).

the film decreases with an increase in annealing temperature. This may be explained as the increase in grain size with an increase in annealing temperature may decrease the surface area of each grain and thereby a reduction in force per unit area between grains and consequently pave the way for strain relaxation [52].

The SEM images of the as-deposited PbS thin films on glass substrate are shown in figure 3. Generally, the overall morphology of the PbS thin films at the three annealing temperatures (250, 300 and 400°C) consist of cubic crystals, with varying degrees of plane-related growth of conjoined interlocking crystals visually evident. The frequency of occurrence of conjoined crystals appeared to vary between the three samples. It was less frequent in the samples at relatively low temperatures (250, 300°C), but very frequent in the sample annealed at high temperature (400°C). The range of crystal dimensions appeared to vary between the annealing temperatures as follows: 1.7–4.4 µm, 1.7–2.3 µm and 1.8–2.0 µm for spin-coated glass substrates annealed at 250, 300 and 400°C, respectively.

EDX spectroscopy of the PbS thin film did differ significantly from the theoretical ratio of 1 : 1 expected for PbS. Generally, the % S decreased slightly as the annealing temperature was increased from 250 to 400°C (figure 4). However, at all cases, the ratio of Pb to S was approximately 1 : 1 as reported elsewhere in the literature [34].

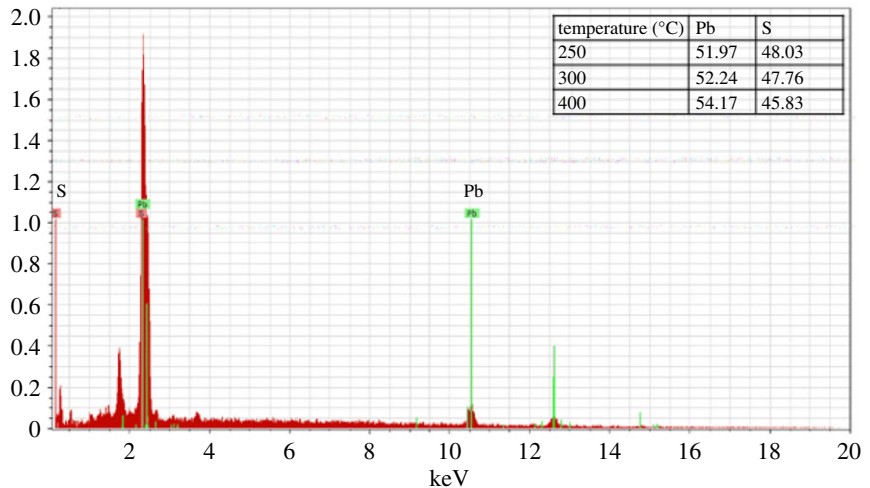

**Figure 4.** EDX spectrum of PbS thin films at 250°C (inset) table of % Pb and % S.

| temperature (°C) | Pb | S |
|---|---|---|
| 250 | 51.97 | 48.03 |
| 300 | 52.24 | 47.76 |
| 400 | 54.17 | 45.83 |

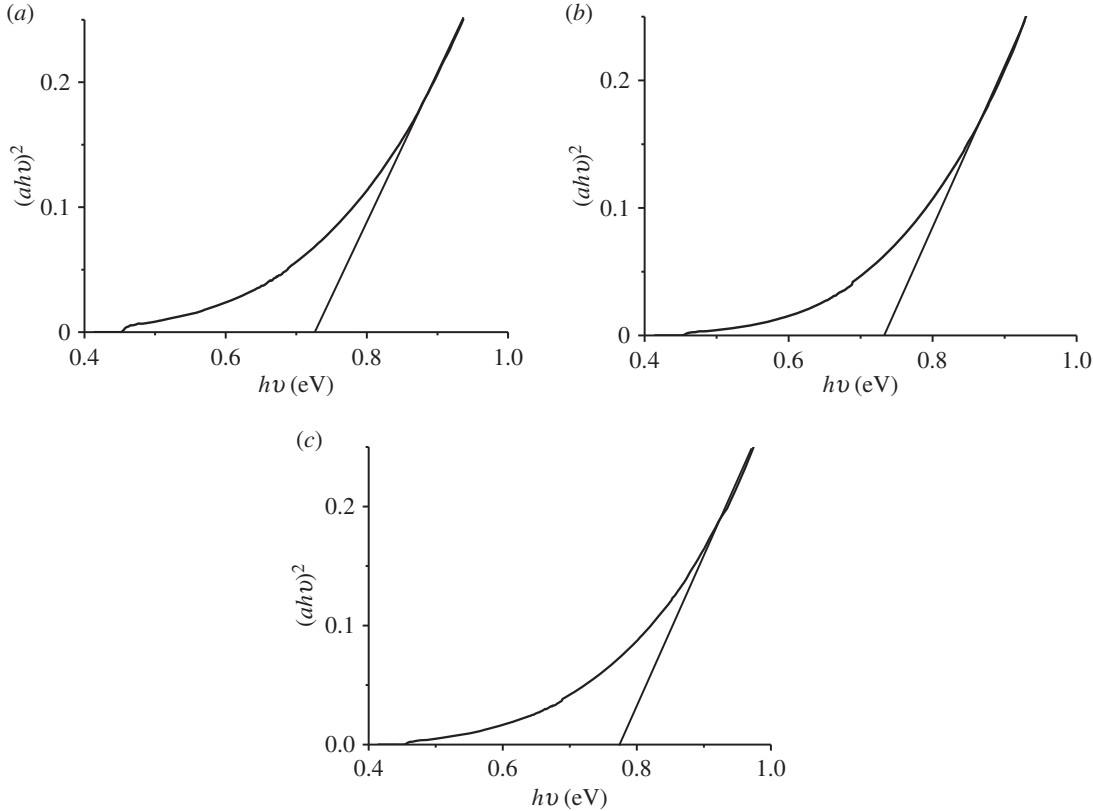

**Figure 5.** Tauc plots of PbS thin films at (a) 250°C, (b) 300°C and (c) 400°C.

PbS is a direct band gap semiconductor and hence a plot of $(\alpha h\upsilon)^2$ against $h\upsilon$ is expected to show a linear portion that corresponds to the energy of the optical band gap when extrapolated to the $h\upsilon$ axis. Optical properties of PbS were estimated using Tauc plots as 0.72, 0.73 and 0.77 eV for thin films annealed at 250, 300 and 400°C (figure 5). These band gaps were blue shifted from the bulk value of 0.41 eV and also conform to earlier reports on the band gap of PbS thin films as reviewed in the literature [21]. From the band gaps obtained, the PbS thin films can be used as acceptors in solar cells.

## 4. Conclusion

Face-centred cubic PbS thin films with the 200 planes as the most preferred orientation have been deposited from lead ethyl carbamate SSP using the spin coating technique followed by annealing at

moderate temperatures. SEM analyses revealed the formation of well-resolved cubes of sizes ranging between 1.7 and 2.3 µm depending on the annealing temperature. Optical band gaps of the PbS thin films were estimated to range from 0.72 to 0.77 eV which has been blue shifted from the bulk band gap.

Data accessibility. Data have been uploaded as part of the supplementary material.

Authors' contributions. S.A.S. synthesized and characterized complexes. N.O.B. analysed and wrote the sections of the XRD. D.A.-P. analysed and wrote the sections of the UV–VIS–NIR and band gap. C.W. analysed and wrote the sections of the SEM and EDX.

Competing interests. We declare we have no competing interests

Funding. We received no funding for this study.

Acknowledgements. The University of Energy and Natural Resources, Sunyani, Kwame Nkrumah University of Science and Technology, Kumasi and The University of Manchester, United Kingdom are acknowledged for providing laboratory space for this research.

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
