## [Reviewer comments · Royal Society Open Science]

Review History

RSOS-190943.R0 (Original submission)

Review form: Reviewer 1

Is the manuscript scientifically sound in its present form?

Yes

Are the interpretations and conclusions justified by the results?

Yes

Is the language acceptable?

No

Is it clear how to access all supporting data?

Yes

Do you have any ethical concerns with this paper?

No

Have you any concerns about statistical analyses in this paper?

No

Recommendation?

Accept with minor revision (please list in comments)

Comments to the Author(s)

In this manuscript, you presented that lead ethyl dithiocarbamates have been successfully used as single source precursors for the deposition of PbS using spin coating followed by annealing at moderate temperatures. Although the method is unique, there are still some shortcomings in the paper. I believe the paper may be accepted for publication after carefully addressing the following points.

1- In your paper, you stated that you have synthesised the Single source precursors, maybe you can characterize it, not by just words.

2-On line 16 of paper 6, in the sentence "The 100% conversion of the complex to PbS results in 44.60 % of the total mass.", "44.6%" did not appear in the corresponding references. Can you explain more clearly?

3-For the SEM images, the structures changing mechanism of PbS nanocrystal with temperature increase is unclear.

4-In tabel 1, the form should be a three-line table.

5-Some writing mistakes need to be corrected. Page 5, line 18: "Melting point 210-211°C" should be "Melting point is 210-211°C".

6- The analysis of XRD data in this paper need to be improved. There are many previous works about the analysis of XRD data, such as "10.1016/j.electacta.2018.10.039", "10.1016/j.snb.2019.02.026" and so on, and the authors could refer them to enrich your argument and increase the persuasiveness of your article.

Review form: Reviewer 2**Is the manuscript scientifically sound in its present form?**

No

Are the interpretations and conclusions justified by the results?

Yes

Is the language acceptable?

Yes

Is it clear how to access all supporting data?

Yes

Do you have any ethical concerns with this paper?

No

Have you any concerns about statistical analyses in this paper?

No

Recommendation?

Major revision is needed (please make suggestions in comments)

Comments to the Author(s)

The work is interesting and they prepared cubic shape PbS particles from the single precursor Lead ethyl dithiocarbamate. Since the work only the synthesis process of PdS particles. So, to improve the quality further of the work, need more characteristic information of the prepared PdS materials. If author replied with the answer to all question, then the paper can be published after careful observation.

1. It is necessary to highlight that the Lead ethyl dithiocarbamate is not yet used by another author to prepare PdS particles by using the same procedure. For this case, the author needs to search for literature and write the asked information.
2. It is requested to provide PL or UV-visible of PdS particles. Based on the PL or UV-visible, the author needs to give some positive discussion point in their articles to highlight the prepared PdS materials for different application.
3. TEM image of PdS is also required.

Decision letter (RSOS-190943.R0)

01-Jul-2019

Dear Dr Saah:

Title: Lead Ethyl Dithiocarbamates: Efficient Single Source Precursors to PbS Nanocubes
Manuscript ID: RSOS-190943

The editor assigned to your manuscript has now received comments from reviewers. We would like you to revise your paper in accordance with the referee and Subject Editor suggestions which can be found below (not including confidential reports to the Editor). Please note this decision does not guarantee eventual acceptance.

Please submit your revised paper before 24-Jul-2019. Please note that the revision deadline will expire at 00.00am on this date. If we do not hear from you within this time then it will be assumed that the paper has been withdrawn. In exceptional circumstances, extensions may be possible if agreed with the Editorial Office in advance. We do not allow multiple rounds of revision so we urge you to make every effort to fully address all of the comments at this stage. If deemed necessary by the Editors, your manuscript will be sent back to one or more of the original reviewers for assessment. If the original reviewers are not available we may invite new reviewers.

Please also include the following statements alongside the other end statements. As we cannot publish your manuscript without these end statements included, if you feel that a given heading is not relevant to your paper, please nevertheless include the heading and explicitly state that it is not relevant to your work.

- Funding statement

Please include a funding section after your main text which lists the source of funding for each author.

RSC Associate Editor:
Comments to the Author:
(There are no comments.)

RSC Subject Editor:
Comments to the Author:
(There are no comments.)

Reviewers' Comments to Author:
Reviewer: 1

Comments to the Author(s)
In this manuscript, you presented that lead ethyl dithiocarbamates have been successfully used as single source precursors for the deposition of PbS using spin coating followed by annealing at moderate temperatures. Although the method is unique, there are still some shortcomings in the

paper. I believe the paper may be accepted for publication after carefully addressing the following points.

1- In your paper, you stated that you have synthesised the Single source precursors, maybe you can characterize it, not by just words.

2-On line 16 of paper 6, in the sentence "The 100% conversion of the complex to PbS results in 44.60 % of the total mass.", "44.6%" did not appear in the corresponding references. Can you explain more clearly?

3-For the SEM images, the structures changing mechanism of PbS nanocrystal with temperature increase is unclear.

4-In tabel 1, the form should be a three-line table.

5-Some writing mistakes need to be corrected. Page 5, line 18: "Melting point 210-211°C" should be "Melting point is 210-211°C".

6- The analysis of XRD data in this paper need to be improved. There are many previous works about the analysis of XRD data, such as "10.1016/j.electacta.2018.10.039" , "10.1016/j.snb.2019.02.026" and so on, and the authors could refer them to enrich your argument and increase the persuasiveness of your article.

Reviewer: 2

Comments to the Author(s)

The work is interesting and they prepared cubic shape PbS particles from the single precursor Lead ethyl dithiocarbamate. Since the work only the synthesis process of PdS particles. So, to improve the quality further of the work, need more characteristic information of the prepared PdS materials. If author replied with the answer to all question, then the paper can be published after careful observation.

1. It is necessary to highlight that the Lead ethyl dithiocarbamate is not yet used by another author to prepare PdS particles by using the same procedure. For this case, the author needs to search for literature and write the asked information.

2. It is requested to provide PL or UV-visible of PdS particles. Based on the PL or UV-visible, the author needs to give some positive discussion point in their articles to highlight the prepared PdS materials for different application.

3. TEM image of PdS is also required.

Author's Response to Decision Letter for (RSOS-190943.R0)

See Appendix A.

RSOS-190943.R1 (Revision)

Review form: Reviewer 2

Is the manuscript scientifically sound in its present form?

Yes

Are the interpretations and conclusions justified by the results?

Yes

Is the language acceptable?

Yes

Do you have any ethical concerns with this paper?

No

Have you any concerns about statistical analyses in this paper?

No

Recommendation?

Accept as is

Comments to the Author(s)

The author replied and provided all of the necessary question and data. So, the paper can be published.

Decision letter (RSOS-190943.R1)

23-Sep-2019

Dear Dr Saah:

Title: Lead Ethyl Dithiocarbamates: Efficient Single Source Precursors to PbS Nanocubes
Manuscript ID: RSOS-190943.R1

It is a pleasure to accept your manuscript in its current form for publication in Royal Society Open Science. The chemistry content of Royal Society Open Science is published in collaboration with the Royal Society of Chemistry.

RSC Associate Editor:
Comments to the Author:
I apologise that this has taken longer than usual.

RSC Subject Editor:
Comments to the Author:
(There are no comments.)

Reviewer(s)' Comments to Author:
Reviewer: 2

Comments to the Author(s)
The author replied and provided all of the necessary question and data. So, the paper can be published.

Appendix A

Lead Ethyl Dithiocarbamates: Efficient Single Source Precursors to PbS Nanocubes

Response to reviewers

Reviewer 1		
Sn	Comment	Response
1.	In your paper, you stated that you have synthesised the Single source precursors, maybe you can characterize it, not by just words.	Although the single crystal has been reported earlier, the complex was characterized by using micro-elemental analyses, infrared spectroscopy, melting point, and ¹ H NMR which were all indicated in the write-up.
2.	On line 16 of paper 6, in the sentence "The 100% conversion of the complex to PbS results in 44.60 % of the total mass.", "44.6%" did not appear in the corresponding references. Can you explain more clearly?	The reference is on the "The difference however between the two values is 4.90% which implies that the compound can be decomposed to produce PbS as residue." The corresponding reference also reported that the % residue of a complex when converted to the metal chalcogen is lower than the % residue obtained from the TGA analyses.
3.	For the SEM images, the structures changing mechanism of PbS nanocrystal with temperature increase is unclear.	As the temperature increases (400 °C), the crystals have high surface energy and therefore they collide faster leading to the formation of conjoined crystals as seen in Figure 3e. This observation is however not prominent at lower temperatures (250 °C)
4.	In table 1, the form should be a three-line table.	Done
5.	Some writing mistakes need to be corrected. Page 5, line 18: "Melting point 210-211°C" should be "Melting point is 210-211°C".	The correction has been done.
6	The analysis of XRD data in this paper need to be improved. There are many previous works about the analysis of XRD data, such as "10.1016/j.electacta.2018.10.039", "10.1016/j.snb.2019.02.026" and so on, and the authors could refer them to enrich your argument and increase the persuasiveness of your article.	The references have been looked at and cited appropriately to enrich the arguments.

Reviewer: 2

1	It is necessary to highlight that the Lead ethyl dithiocarbamate is not yet used by another author to prepare PdS particles by using the same procedure. For this case, the author needs to search for literature and write the asked information.	Literature has been sought and publications that involve the use of lead diethyldithiocarbamates have been cited appropriately. This would however, be the first publication of the use of lead diethyldithiocarbamate in the syntheses of PbS thin films using the spin coating method.
2	It is requested to provide PL or UV-visible of PdS particles. Based on the PL or UV-visible, the author needs to give some positive discussion point in their articles to highlight the prepared PdS materials for different application.	The UV spectra have been provided in the supplementary data. Also, highlights on the potential application of the as-synthesized PbS thin films have been included in the manuscript.
3	TEM image of PdS is also required.	Considering the sizes of the particles, the SEM images are more appropriate as compared to the TEM. We shall consider this suggestion in future works.